# The Association between Adverse Childhood and Adulthood Experiences, Social Isolation, Loneliness, and Depression among Young Adults in South Korea

**DOI:** 10.3390/ijerph20196900

**Published:** 2023-10-09

**Authors:** Sung Man Bae

**Affiliations:** Department of Psychology and Psychotherapy, Dankook University, Cheonan 31116, Republic of Korea; spirit73@dankook.ac.kr

**Keywords:** young adult, depressive symptoms, adverse childhood experiences, adverse adulthood experiences, social isolation, loneliness

## Abstract

This study examined the association between social isolation, loneliness, and depressive symptoms among young adults in South Korea after controlling for sociodemographic variables, adverse child and adult experiences, and perceived health status. The researcher analyzed the data of 2014 young adults aged 18–34 years from the Youth Social Economic Survey using hierarchical multiple regression analysis conducted with IBM SPSS Statistics 28.0. Among the sociodemographic variables, sex, education, and household income were significantly related to depressive symptoms. Females reported higher levels of depressive symptoms than males, and those attending college or having a college (or higher) degree reported lower levels of depressive symptoms than those with a high-school diploma or lower. Higher household income was associated with lower levels of depressive symptoms. Perceived health status was negatively associated with depressive symptoms, while adverse childhood and adulthood experiences, social isolation, and loneliness were positively associated with it. Among the major independent variables, loneliness was most strongly related to depressive symptoms, whereas social isolation had the weakest relevance to it. The prediction model proposed explained 32.7% of the variance in depressive symptoms and was considered good. Therefore, focusing on loneliness may be more important than focusing on social isolation to prevent and deal with depressive symptoms among young adults.

## 1. Introduction

Social isolation has been treated as a social problem for older adults. However, with individualization, the weakening of community, and the acceleration of Information and Communications Technology (ICT), the number of young people becoming socially isolated has been increasing rapidly. According to the 2021 Youth Social and Economic Status Survey targeting young Koreans (18–34 years old), 35.0% of 2014 young people experience social isolation to some degree in their daily lives [1]. In addition, 13.4% of the participants reported high internal social isolation (loneliness), and according to the Organization for Economic Co-operation and Development (OECD), 21.59% of young Koreans are NEETs (not in education, employment, or training)—a situation similar to social isolation [2]. Several studies have identified that the social isolation and loneliness experienced by older adults have a significant effect on their mental health, especially depressive symptoms [3,4]. However, as studies on young adults are scarce, additional research is required [5].

Depressive symptoms, such as depressed mood, decreased interest, and lethargy, are some of the representative emotional problems of young adults and are an important factor in determining their academic achievement, job performance, and interpersonal functioning. Depression is the third most important factor causing disease worldwide, and in particular, it has been reported that approximately 40% of people under the age of 20 experience their first depressive episode, and the prevalence of depression peaks in the second and third decades [5,6].

Social isolation and loneliness share the concept of weak social relationships but are distinct concepts [7]. Social isolation refers to a state in which social networks are limited or absent. In contrast, loneliness refers to the subjective lack of social interaction [8], which is the emotional discomfort felt when a discrepancy occurs between desired social connections and actual social interactions [9,10,11,12]. Social isolation is closely related to the quantity and frequency of social contact [13]. In contrast, loneliness is associated with the quality of social relationships [14].

Several studies on older adults have identified relationships between social isolation, loneliness, and depressive symptoms [13,15,16]. Loneliness, in particular, was strongly related to mental illness, especially depressive symptoms [17,18,19]. In a longitudinal study by Zhang et al. [20] involving 634 older adults in Shanghai, moderate-to-severe loneliness was associated with more severe depressive symptoms, and persistent social isolation was associated with stronger depressive symptoms. In a longitudinal study by Santini et al. [21] on adults aged 50 years in Ireland, poor social contact and loneliness significantly increased depressive and anxiety symptoms. However, few studies on young adults have shown differences in the degree of influence of social isolation and loneliness on depressive symptoms. In a longitudinal study of 2232 young British adults, social isolation and loneliness were positively related to depressive symptoms. Loneliness was more strongly related to depressive symptoms when both variables were simultaneously input into the regression model at the same time [22]. In a study of Korean adults aged 15–74 years conducted by Kim et al. [13], 12.9% of the socially isolated group and 39.1% of the lonely group were diagnosed with depressive symptoms (PHQ-9 > 10).

Social isolation does not necessarily cause loneliness or depressive symptoms [8]. Although some studies have suggested that more social contact may prevent loneliness and mental health problems [13], people with fewer social contacts do not necessarily experience loneliness [23]. These results imply that experiencing emotional intimacy and connection with close people is more important than the size of social networks or frequency of social contact, and that the mere absence of physical contact with others does not cause loneliness and depressive symptoms [24].

To analyze the association between social isolation, loneliness, and depressive symptoms more accurately, it is necessary to control for covariates related to these variables. Adverse childhood experiences (ACEs) are closely associated with social isolation, loneliness, and depressive symptoms [25,26]. ACEs refer to experiences of various adversities in childhood, including abuse and neglect (physical/emotional/sexual abuse, physical/emotional neglect) and family dysfunction (parental divorce, domestic violence, etc.). In a study of a UK population cohort of children born in 1958, it was found that experiencing ACEs (parental neglect, alcoholism, and criminality) between the ages of 7 and 16 years strongly predicted mood problems up to the age of 50 [27]. Based on past studies, both ACEs and adverse adulthood experiences (AAEs, e.g., failure to find a job, workplace maladjustment, and interpersonal difficulties) may be closely related to social isolation and depressive symptoms. However, most studies have not considered ACEs and AAEs simultaneously.

Perceived health status has also been shown to be closely related to depressive symptoms in several studies involving young and older adults [28]. In a study by Liu et al. [29] on 6485 Chinese older adults, perceived health status had a negative effect on depressive symptoms, even after controlling for age, sex, marital status, education, and residency type. In a study by Lin, Hung, and Yang [30] on 700 college students, sex and grade were controlled, and perceived health status was negatively related to depressive symptoms.

Among the sociodemographic variables, sex is closely related to depressive symptoms, and previous studies have reported higher levels of depressive symptoms in females than in males [10,15]. Several studies have verified that education and economic status are negatively related to depressive symptoms [29,31]. Age has been found to be positively related to depressive symptoms in some studies [31], but the relationship between these two variables is relatively unclear.

In summary, social isolation and loneliness appear to be important variables for depressive symptoms not only in older adults but also in young adults. However, most previous studies have been conducted on older adults, and many have not controlled for covariates such as ACEs, AAEs, and perceived health status, which are closely related to social isolation, loneliness, and depressive symptoms. In fact, in a study by Hsiao Peng et al. [32] on adults aged 50 years and older, the relationship between social isolation and depressive symptoms was not significant after controlling for sex, marital status, and self-reported health.

Another important issue is that of the representative scales of social isolation (e.g., the Lubben Social Network Scale-6), which measure the frequency of interactions with family, friends, and acquaintances on offline networks but not on online networks. However, with the development of ICT and smartphones, online and virtual-world interactions have accounted for a large portion of social contact. Shaw and Gant [33] verified that online and virtual interactions significantly reduce loneliness and depressive symptoms; moreover, online interactions contribute to gaining social support and maintaining a sense of belonging [34]. Despite the importance of online social contact, most studies have not included online interactions when measuring social isolation. Therefore, this study measured social isolation more broadly by including online social contacts.

This study aimed to verify the association between social isolation, loneliness, and depressive symptoms in young adults by controlling for sociodemographic variables, ACEs, AAEs, and perceived health.

## 2. Materials and Methods

### 2.1. Survey and Participants

This study analyzed data from the Youth Social Economic Survey conducted by the National Youth Policy Institute (NYPI). The survey targeted a sample of approximately 2000 young people aged 18–34 years residing in general households in South Korea. Population and housing census survey district lists were used as the sampling framework. The sample design comprised a two-step stratification process: in the first step, stratification was performed at the level of 16 provinces (special and metropolitan cities): 7; provinces: 9; in the second step, the nine provinces were stratified into dong, eup, and myeon. For the sampling method, probability proportional to size systematic sampling was used for each floor of the sampling district in the first round. In the second round, households were extracted through systematic sampling in each sampling district (Figure 1). The survey period lasted approximately 3 months (from 19 July to 31 October 2021), and trained professional interviewers conducted individual visits and interviews. The main interview method was the Tablet PC Aided Personal Interview (TAPI). All research procedures in this study were approved by the Institutional Review Board of Dankook University (approval number: 2023-08-012-002).

### 2.2. Measures

#### 2.2.1. Depressive Symptoms

The Korean version of the CES-D-10 (Boston form) was used to assess depressive symptoms. The CES-D-10 is an abbreviated version of the CES-D developed by Radloff [35]. The scale consists of 10 items, and each item is rated on a 4-point Likert scale (0 = very rare, less than 1 day/week; 1 = sometimes, 1–2 days/week; 2 = often, 3–4 days/week; 3 = most of the time, 5 or more days/week); the higher the total score, the more severe the depressive symptoms. Examples of representative questions are as follows. (1) I felt down, depressed, or hopeless. (2) I lost interest in or felt no enjoyment in the things I used to do. (3) Difficulty falling asleep or waking up frequently/or sleeping too much. The Cronbach’s α was 0.850 in this study.

#### 2.2.2. Social Isolation

To measure social isolation, the research team modified and used the Korean version of the Lubben Social Network Scale-18 (LSNS-18), adapted and validated by Kim [36] and developed by Lubben et al. [37], which measures the frequency of face-to-face and non-face-to-face contact with the main objects (family, relatives, close friends, people you know at work/school/neighborhood, and people you know through online communities). This is a 6-point Likert scale (6 = almost every day, 5 = once or twice a week, 4 = once or twice a month, 3 = once or twice a month, 2 = once or twice a year, and 1 = no interaction). All items were used for reverse scoring. The scale consists of 10 items (five face-to-face contacts and five non-face-to-face contacts), and the higher the total score, the higher the degree of social isolation. The Cronbach’s α for this study was 0.735.

#### 2.2.3. Loneliness

To measure loneliness, the researcher used a scale modified and validated by Jin and Hwang [38] after consultation with experts and based on the UCLA Loneliness Scale (Version 3) developed by Russell, Peplau, and Ferguson [39] and revised by Russell [40]. Examples of representative questions are as follows: (1) I feel connected with people around me; (2) I feel that my friendships are lacking; (4) I feel alone; (9) I feel close to people; (10) I feel alienated; (11) I feel my relationships with others are meaningless; (18) I feel isolated from others. This scale consisted of 18 items on a 4-point scale (1 = never, 2 = rarely, 3 = sometimes, 4 = always), and the higher the score, the stronger the loneliness experience. The Cronbach’s α for this scale in this study was 0.860.

#### 2.2.4. Perceived Health Status

In this study, a single item on a 5-point Likert scale (1 = not at all, 2 = disagree, 3 = average, 4 = agree, and 5 = strongly agree) was used to measure perceived physical health status. Participants had to answer the question, “Do you consider yourself physically fit?”.

#### 2.2.5. Adverse Childhood Experiences (ACEs)

To measure ACEs, the research team used a modified and supplemented scale based on the Adverse Childhood Experience scale translated by Ryu et al. [41], the Adverse Childhood Experiences International Questionnaire (ACE-IQ) developed by the World Health Organization [42], and the Factual Survey of Gwangju Metropolitan City Recruit by Lim et al. [43]. It consisted of six questions, to which the participants had to answer “yes” or “no”. The specific types of ACEs measured by the scale are as follows: (1) experience of rapidly becoming difficult family circumstances; (2) experience of excessive corporal punishment or emotional attack from a caregiver; (3) experience of having a family member with emotional problems; (4) experience of losing a close person; (5) experience of changing schools or moving frequently; and (6) school experience of being harassed or bullied by others in my neighborhood. The higher the score, the greater the number of ACEs.

#### 2.2.6. Adverse Adulthood Experiences (AAEs)

To measure AAEs, the research team used five items, to which the participants had to answer “yes” or “no”. The specific types of AAEs were as follows: (1) experience of having to take a leave of absence or drop out of school due to financial difficulties; (2) experience of being pressured to find a job by someone close to me or being forced to change career paths; (3) not being able to enter college at the time I wanted; (4) not being able to find a job at the time I wanted; and (5) experiences of being betrayed or scammed by someone I trusted. The higher the score, the higher the AAEs.

#### 2.2.7. Demographic Variables

In this study, sex (male = 0, female = 1), age, education (high-school diploma or below = 0, college attending/degree or higher = 1), and household income (range = 1–10) were included as covariates in the multiple regression model.

### 2.3. Analysis

IBM SPSS Statistics version 28.0 was used for data analysis. First, Cronbach’s α was calculated to verify the reliability of the measures, and frequency analysis was conducted to identify the demographic characteristics of the participants. Second, a descriptive statistical analysis was conducted to identify the mean, standard deviation, skewness, and kurtosis of the main variables. Pearson’s correlation analysis was performed to verify the correlation between the main variables. Finally, hierarchical multiple regression analysis was conducted to verify the association between the main variables.

## 3. Results

### 3.1. Descriptive Statistics for Sociodemographic Variables

Table 1 presents specific information on the sociodemographic variables. A total of 2041 participants (1074 male [52.6%]; 967 female [47.4%]) were included in this study. The average age was 26.21 years (SD = 4.69, range = 18–34). A total of 1573 people attended/graduated college or had a higher education level, accounting for 77.1% of the sample; 468 people had a high-school diploma or lower education level, accounting for 22.9%.

### 3.2. Correlation

The results of the correlation analysis between the main variables are presented in Table 2. Perceived health status is negatively correlated with ACEs, AAEs, social isolation, loneliness, and depressive symptoms. ACEs were positively correlated with AAEs, social isolation, loneliness, and depressive symptoms. AAEs were positively correlated with social isolation, loneliness, and depressive symptoms. Social isolation was positively correlated with loneliness and depressive symptoms. Finally, loneliness was positively correlated with depressive symptoms.

### 3.3. Hierarchical Multiple Regression

The results of the hierarchical multiple regression analysis are presented in Table 3. Among sociodemographic variables, sex was positively related to depressive symptoms in young adults. In other words, young female adults reported higher levels of depressive symptoms than male adults. Education level was also negatively associated with depressive symptoms. Those attending/having graduated college or with a higher degree reported lower levels of depressive symptoms than those with a high-school diploma or lower education level. Household income was negatively related to depressive symptoms, and young adults with higher household incomes reported lower depressive symptoms than those with lower household incomes. Finally, age was not significantly associated with depressive symptoms. Sociodemographic variables accounted for 3.6% of the total variance in depressive symptoms.

Perceived health status was negatively related to depressive symptoms in young adults, indicating that a more positive perception of one’s own health status is related to lower depressive symptoms. ACEs and AAEs were positively associated with depressive symptoms, indicating that higher numbers of ACEs and AAEs are related to higher levels of depressive symptoms. Social isolation and loneliness were also positively related to depressive symptoms, indicating that stronger social isolation and loneliness are related to higher levels of depressive symptoms. Among the main independent variables, loneliness was most strongly associated with depressive symptoms in young adults, while social isolation was weakly associated with depressive symptoms. The total explanatory value of the independent variables included in the final regression model was 32.7%, and the prediction model for depressive symptoms proposed in this study was judged to be good.

## 4. Discussion

This study aimed to verify the relationship between social isolation, loneliness, and depressive symptoms in young adults after controlling for sociodemographic variables, perceived health, ACEs, and AAEs. A discussion of the main results follows.

ACEs were positively related to depressive symptoms in young adults, consistent with previous studies [44,45]. Evidence suggests that ACEs increase responsiveness to psychosocial stress [46]; specifically, those who have experienced child abuse have higher adrenocorticotropic hormone (ACTH), cortisol levels, and heart rate responses to stress, and these tendencies persist into adulthood, amplifying the influence of stress on depressive symptoms [47]. AAEs were also positively correlated with depressive symptoms. A strong proximal stressful event is a representative risk factor for depressive symptoms; in this respect, AAEs can be major predictors of adult depressive symptoms. This study suggests that, similar to ACEs, AAEs may be important in the prediction and prevention of depressive symptoms in young adults [48].

Social isolation was positively associated with depressive symptoms, consistent with the findings of previous studies. Importantly, among the main variables included in the predictive model, social isolation was weakly related to depressive symptoms. This study suggests that the degree of relevance of social isolation and depressive symptoms in young adults may be significantly lower than that in older adults. Older adults experiencing social isolation are also more likely to experience loneliness and depressive symptoms. Social isolation due to the environment of old age, such as retirement and the death of loved ones, is likely to cause a lack of a sense of belonging and a burden on others; as a result, older adults are more likely to experience loneliness and depressive symptoms [49]. However, young adults who experience social isolation may not necessarily experience loneliness or depressive symptoms.

Stronger loneliness is associated with higher levels of depressive symptoms. Among the independent variables, loneliness had the strongest association with depressive symptoms. This study clarified that loneliness is a key variable in predicting, preventing, and intervening in depressive symptoms among young adults. Recent studies exploring the fact that loneliness and depressive symptoms share biomarkers such as inflammation, high-sensitivity C-reactive protein, and bilirubin support a strong association between the two variables [50,51]. Young adults are more active in social activities and interpersonal contact than older adults. More importantly, loneliness may be experienced even if the frequency of social contact is high. Similarly, more social contact does not completely protect against loneliness, and less social contact does not necessarily cause loneliness or depressive symptoms [11,23].

The results of this study imply that it is necessary to focus more on loneliness experiences in preventing and intervening in depressive symptoms among young adults and that the composition and functioning of social networks may be more important than the size of one’s social network [11]. In addition, the results suggest that social isolation and loneliness need to be comprehensively considered to prevent and effectively intervene in depressive symptoms among young people and that effective intervention strategies need to be developed and applied to alleviate loneliness [22]. Ji, Basanovic, and MacLeod [52] argue that social activities could promote resilience against loneliness in individuals with depressive symptoms. Arnold and Winkielman [53] suggest that loneliness is related to impaired spontaneous smile mimicry and that it is advised to smile (but only deliberately) even though one is suffering.

Among sociodemographic variables, sex, education, and family income were significantly related to depressive symptoms. Females reported more depressive symptoms than males, consistent with previous findings. Higher household income was associated with lower depressive symptoms. In Korean society, where complete economic independence is difficult in young adulthood, the higher economic level of the parents or family may reduce daily stress and depressive symptoms among young people; this is in line with previous studies showing that people with a college degree or higher reported lower levels of depressive symptoms than those with a high-school degree or lower. 

The contributions and implications of this study are as follows: First, while previous studies exploring the relevance of social isolation, loneliness, and depressive symptoms have mainly focused on older adults, this study verified the association between these variables in a young adult group. In particular, our study controlled for ACEs, AAEs, and perceived health status as covariates, which have been overlooked in previous studies. Second, unlike previous studies, this study broadly measured adverse life experiences by adding AAEs and more accurately measured social isolation by adding online social contact. These efforts can contribute to verifying the relationships between social isolation, loneliness, and depressive symptoms. The results of this study imply that efforts to prevent and decrease loneliness, which reflects the qualitative aspect of social relationships, may be more important than focusing on social isolation, which reflects objective social networks, in preventing and reducing depressive symptoms in this group.

### 4.1. Limitations

The limitations of this study and suggestions for future research are as follows. First, as this is a cross-sectional study, it cannot fully explain the causal relationships between the variables. Second, as it targeted young adults between the ages of 18 and 34 years, the results cannot be generalized to all adults. 

### 4.2. Recommendations for Future Research

Few studies have verified the relationships between the variables in more detail. Therefore, in future studies, it will be necessary to make efforts to verify the specific relationships between variables by targeting various age groups—for example, through a model in which loneliness mediates the relationship between social isolation and depressive symptoms. Because this study analyzed the data of cross-sectional design, it is difficult to clarify causality between variables. Therefore, reverification through longitudinal design is necessary to clarify further the influence of social isolation and loneliness on youth depression.

## 5. Conclusions

The results of this study suggest that social isolation and loneliness are important predictors of depressive symptoms not only in older adults but also in the early adulthood group. In particular, loneliness, a qualitative aspect of social contact, was more strongly associated with depressive symptoms in early adulthood than social isolation, a quantitative aspect of social contact. These results suggest that efforts to develop and apply strategies to reduce loneliness are needed to prevent and intervene in depressive symptoms in early adulthood.

## Figures and Tables

**Figure 1 ijerph-20-06900-f001:**
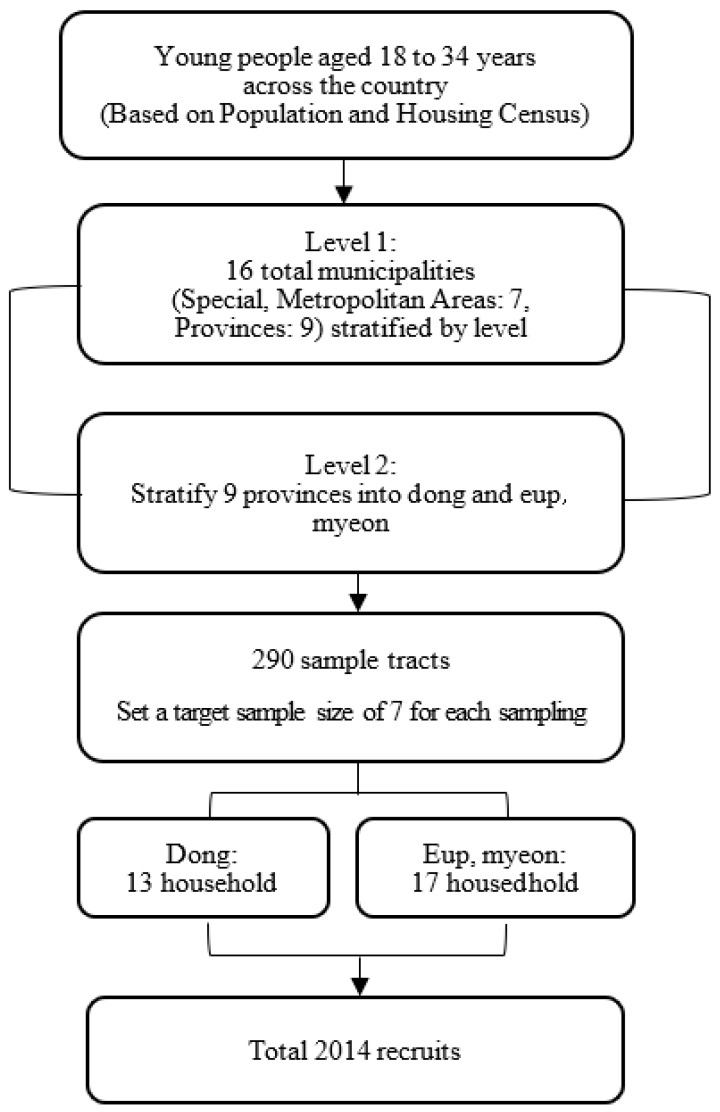
Flowchart for the entire selection procedure.

**Table 1 ijerph-20-06900-t001:** General characteristics of included studies (*n* = 2041).

	*n*	%
**Sex**		
Male	1074	52.6
Female	967	47.4
**Age group**		
18–19 years	185	9.1
20–24 years	588	28.8
25–29 years	668	32.7
30–34 years	600	29.4
**Age**		
Mean = 26.21, SD = 4.69, Range = 18–34		
**Education**		
Under high-school graduation	468	22.9
A college degree or higher	1573	77.1
**Family Income (FI)**		
1	15	0.7
2	26	1.3
3	76	3.7
4	168	8.2
5	497	24.4
6	508	24.9
7	520	25.5
8	203	9.9
9	23	1.1
10	5	0.2
**Marital status**		
Married	345	16.9
Widowed or divorced	5	0.2
Single	1691	82.9

**Table 2 ijerph-20-06900-t002:** Descriptive statistics and correlation (*n* = 2041).

Variables	1	2	3	4	5	6
1. PHS						
2. ACEs	−0.16 ***					
3. AAEs	−0.12 ***	0.44 ***				
4. SI	−0.17 ***	12 ***	0.06 **			
5. L	−0.31 ***	0.19 ***	0.13 ***	0.20 ***		
6. D	−0.34 ***	0.33 ***	0.28 ***	0.17 ***	0.47 ***	
Mean	4.05	0.69	0.45	29.69	32.99	3.38
SD	0.69	1.27	0.88	7.41	7.31	4.22
Skewness	−0.97	2.27	2.48	0.20	0.40	1.99
Kurtosis	2.60	5.27	7.08	0.01	0.26	5.20

Note. ** *p* < 0.01, *** *p* < 0.001.

**Table 3 ijerph-20-06900-t003:** Hierarchical multiple regression for predicting depressive symptoms (*n* = 2041).

	Variables	B	SE	β	*t*
Step 1	Sex	0.563	0.181	0.071 **	3.107
	Age	0.022	0.024	0.023	0.909
	Edu	−0.030	0.107	−0.007	−0.281
	FI	−0.467	0.063	−0.182 ***	−7.472
*R*^2^ = 0.034					
Step 2	Sex	0.257	0.166	0.032	1.545
	Age	−0.007	0.022	−0.008	−0.338
	Edu	0.055	0.098	0.013	0.559
	FI	−0.147	0.059	−0.054 *	−2.486
	AA	0.683	0.101	0.157 ***	6.791
	ACEs	0.646	0.082	0.186 ***	7.918
	PH	−1.567	0.121	−0.278 ***	−12.966
*R*^2^ = 0.202					
Step 3	Sex	0.375	0.153	0.047 *	2.454
	Age	−0.024	0.020	−0.025	−1.204
	Edu	0.156	0.090	0.037	1.736
	FI	−0.104	0.054	−0.038	−1.904
	AAEs	0.621	0.092	0.143 ***	6.723
	ACEs	0.503	0.075	0.145 ***	6.678
	PH	−0.947	0.116	−0.168 ***	−8.154
	SI	−0.020	0.010	−0.038	−1.917
	L	0.200	0.011	0.371 ***	18.080
*R*^2^ = 0.329					

* *p* < 0.05, ** *p* < 0.01, *** *p* < 0.001. Edu = Education, FI = Family Income, PH = Perceived Health, ACEs = Adverse Childhood Experience, AAEs = Adverse Adults Experience, SI = Social Isolation, L = Loneliness.

## Data Availability

Not applicable.

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
