# Peer review of "The Association between Adverse Childhood and Adulthood Experiences, Social Isolation, Loneliness, and Depression among Young Adults in South Korea"

_ijerph, 2023, doi:10.3390/ijerph20196900_

Round 1
Reviewer 1 Report
See attached file

A missing word in the first sentence impacted the first paragraph of the Introduction. A few sentence structure issues.
Author Response
Reviewer 1
Question 1: The first sentence is missing a word, which impacts the entire introduction.
Answer 1: I changed it as follows.
Social isolation has been treated as a social problem of older adults.
Question 2: The Introduction begins with many statistics, which again is overwhelming to the reader – also related to the missing word in the first sentence.
Answer 2: I added the missing words.
Question 3: Review the sentence structure for the first sentence on page 3.
Answer 3: I reviewed and corrected the first sentence on page 3
Question 4: Wondered why the gap of the research from the survey of participants – July 19 to Oct 31, 2021
Answer 4: I understand that the survey took about 3 months because it was conducted face-to-face during the COVID-19 period.
Question 5: Measures – each measure had a different tool, which were explained by name and a sentence or two explanations. Seven tools were reported one after another – may think about placing samples of each of the measurement tools, to better enhance understanding of each measurement.
Answer 5: I placed samples of each of the measures to better enhance understanding of each measurement.
Question 6: Question 7: May want to separate and expand the Limitations and Recommendations for future research section.
Answer 6: I separated and expanded limitations and recommendations for future research section.
Reviewer 2 Report
Thank you for the opportunity to review your article. I want to start by saying that I found it to be very well-done. It flowed nicely, and it was very understandable from start to finish. You did a great job with reviewing the data and discussing the conclusions derived from your statistical analyses.
I don't have a lot to add in terms of what I would like to see changed before going up for publication. Here are a couple of things I thought of when reading your article:
I. Delete the symbol you have in line 26 of your paper.
2. I'm wondering how COVID may have been a factor in the data. I noticed that the information was collected from July 19 to October 31, 2021. I am from the United States, so I don't know how South Korea was at that time. Here in the US, there were still many restrictions in place for COVID, including mostly Zoom options for meetings, etc. In a lot of ways, many adults found themselves still socially isolated due to changes from COVID. I would be really curious to see how social isolation would impact depressive symptoms in young adults further removed from COVID.
Author Response
Reviewer 2
Question 1: Delete the symbol you have in line 26 of your paper.
Answer 1: I deleted the symbol.
Question 2: I'm wondering how COVID may have been a factor in the data. I noticed that the information was collected from July 19 to October 31, 2021. I am from the United States, so I don't know how South Korea was at that time. Here in the US, there were still many restrictions in place for COVID, including mostly Zoom options for meetings, etc. In a lot of ways, many adults found themselves still socially isolated due to changes from COVID. I would be really curious to see how social isolation would impact depressive symptoms in young adults further removed from COVID.
Answer 2: At the time of the survey, Korea was also greatly affected by COVID-19. For this reason, I understand that the survey took more than three months. As pointed out, I think COVID-19 may have an impact on social isolation. However, there is no information related to this in the secondary data analyzed in this study. Considering the situation of COVID-19, the influence of social isolation on depression shown in this study may have been overestimated, and there is a need to reconfirm the independent influence of social isolation on depression in the future while excluding the influence of COVID-19
Reviewer 3 Report
Thank you for your opportunity to review an interesting topic, namely, The association between adverse childhood and adulthood experiences, social isolation, loneliness, and depression among young adults in South Korea.
My observations would be as follows:
Information on the significance and incidence of depressive symptoms is missing in the introduction section.
How was the representative sample size calculated?
I suggest that the authors could submit the entire selection procedure in a form of flowchart.
You have only 31 references, but you write about more than 52 sources of literature. References are not correct.
The term “depression” should be replaced by the term “depressive symtoms” throughout the manuscript text.
Throughout the manuscript text, the terms need to be harmonized and only “sex”, “male”, and “female” should be written.
In Table 1, I propose to classify age into groups and include data on the economic status as well as family status of study participants.
Kind regards
Minor editing of English language required.
Author Response
Reviewer 3
Question 1: Information on the significance and incidence of depressive symptoms is missing in the introduction section.
Answer 1: I added the significance and incidence of depressive symptoms in the introduction section.
Question 2: How was the representative sample size calculated?
Answer 2: As this study analyzed secondary data, specific details about the sample size calculation process were not provided. If necessary, I will check with the person in charge.
Question 3: I suggest that the authors could submit the entire selection procedure in a form of flowchart.
Answer 3: I added the entire selection procedure in a form of flowchart.
Question 4: You have only 31 references, but you write about more than 52 sources of literature. References are not correct.
Answer 4: I corrected references in text and references area.
Question 5: The term “depression” should be replaced by the term “depressive symptoms” throughout the manuscript text.
Answer 5: I changed the term “depression” as “depressive symptoms”
Question 6: Throughout the manuscript text, the terms need to be harmonized and only “sex”, “male”, and “female” should be written.
Answer 6: I changed “gender, men, women” as “sex, male, female”
Question 7: In Table 1, I propose to classify age into groups and include data on the economic status as well as family status of study participants.
Answer 7: I classified age into 4 groups and added family status.